# Enhancing Adversarial Transferability in Vision-Language Models via Search-Space Expansion

## Abstract

Adversarial attacks are crucial for evaluating the robustness of vision-language pre-trained (VLP) models. However, existing methods suffer from limited transferability across unseen models, limiting their effectiveness as a universal robustness probe. We attribute this partially to the narrow search space of adversarial examples, which can trap optimization in local optima and lead to overfitting. To address this, we propose SEA (**S**earch-space **E**xpansion **A**ttack), a unified framework that improves cross-model transferability by enlarging the adversarial search space across both modalities. For images, SEA leverages historical updates to explore novel optimization directions, effectively avoiding suboptimal optimization trajectories and overfitting. For text, SEA considers both individual word importance and word interactions, recognizing that less salient words can sometimes yield stronger and more transferable attacks. It performs word substitutions across multiple influential positions rather than focusing solely on the most salient word. Consequently, SEA can substantially disrupt cross-modal interactions across different models. Extensive experiments on diverse benchmarks, VLP models and tasks, supported by rigorous theoretical analysis, demonstrate that SEA significantly advances the state of the art. The source code is provided in the supplementary material.

## 1 Introduction

Vision–language pre-trained (VLP) models leverage large-scale self-supervised learning to capture cross-modal correlations and transferable representations, thereby reducing reliance on labeled data. This capability has established them as foundational backbones for a wide range of downstream tasks (Radford et al., 2021; Li et al., 2022; 2021; Yang et al., 2022). However, real-world applications are often confronted with uncertainties and potential threats, which pose significant security risks to VLP models. Therefore, evaluating their robustness is crucial to ensure reliable performance in practical scenarios.

Adversarial attacks are widely used to evaluate model vulnerability by introducing imperceptible perturbations that mislead predictions (Szegedy et al., 2014; Madry et al., 2018; Moosavi-Dezfooli et al., 2017; Qin et al., 2022). These attacks typically exploit models' reliance on spurious correlations, which are the features used for prediction but not truly relevant to the semantic content of the data (Zhang et al., 2021b; Ilyas et al., 2019; Hendrycks et al., 2021; Qin et al., 2022). Recent studies have shown that VLP models are also susceptible to adversarial attacks, where adversarial examples are generated to have a maximal distance from the original data in the feature space (Zhang et al., 2022; Lu et al., 2023; He et al., 2023; Yin et al., 2023; Zhang et al., 2024).

To fully assess both the capabilities and vulnerabilities of VLP models, a key question remains underexplored: *Do generic, model-agnostic vulnerabilities exist across architectures and objectives?* We posit that adversarial transferability offers the key pathway to study such shared weakness. Moreover, generating model-specific adversarial examples is computationally prohibitive, making highly transferable attacks essential for scalable robustness evaluation. However, achieving transferability in attacks for VLP models is particularly challenging. Differences in architectures and training objectives cause models to rely on distinct, model-specific features when associating im-

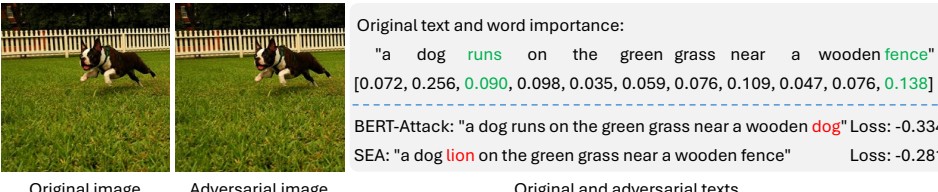

Figure 1: Visualization of adversarial attacks on image and text, and a comparison between our method and BERT-Attack. BERT-Attack targets the second-most important word for replacement, while our method substitutes a less important word. Despite this, our approach achieves a larger separation from the image in the feature space (where a larger loss indicates lower similarity). (Note that the most important word is not replaced, as its available substitute is identical to the original.)

ages and text. In generating adversarial images, commonly used gradient-based methods tend to overfit to the target model, resulting in local optima. To handle this, data augmentation is commonly adopted to regularize the update directions and explore the cross-modal interactions by increasing data diversity and serving as a form of model augmentation. (Lu et al., 2023; Zhang et al., 2024; He et al., 2023; Gao et al., 2024; Jia et al., 2025). While effective, such methods remain fundamentally constrained to the current adversarial trajectory. That is, each update is based on the preceding adversarial example and its gradient, causing successive updates to remain correlated along the same path. Concurrent work (Jia et al., 2025; Gao et al., 2024) attempt to address this limitation by learning gradients from samples selected within an adversarial triangle. However, these methods are restricted to a limited selection region and do not consider diverse update information for learning. As a result, the optimization explores only a narrow set of directions, limiting the search space for transferable adversarial examples. A similar issue arises in the text modality. Current approaches typically perform word substitution on the most important word, identified by masking it and measuring its influence (Lu et al., 2023; Yin et al., 2023; Jia et al., 2025). Nevertheless, replacing only the most salient word does not necessarily yield the most effective attack (As shown in the Figure 1). The effectiveness of a word substitution depends on multiple factors, including the importance of the original word, the choice of replacement, and interactions among words. This highlights the need for a more comprehensive method that can explore a broader adversarial space to enhance adversarial transferability.

In this paper, we propose a simple yet highly effective attack method, termed Search-space Expansion Attack (SEA), which aims to learn adversarial examples within a broader search space. For the image modality, SEA avoids following original optimization trajectories and getting trapped in local optima by exploring diverse search directions using only historical update information. To this end, we introduce two forms of exploration: (1) *gradient-based exploration*, and (2) *perturbation-based exploration*. Beyond extensive empirical evaluation, we provide theoretical analysis to demonstrate its effectiveness from a complementary perspective. In the text modality, instead of restricting substitutions to the most salient word, SEA selects replacements by extending the search to more influential positions, thereby more effectively exploring the influence of candidate substitutions and word interactions. It also balances effectiveness and efficiency, making it practical for large-scale attacks. Extensive experiments demonstrate that SEA consistently outperforms state-of-the-art techniques, significantly improving adversarial transferability across both image and text modalities.

## 2 METHODOLOGY

### 2.1 PRELIMINARIES

The proposed method is developed under a black-box setting, particularly for cross-model scenarios where the target model and tasks are unknown. To address this challenge, we adopt a transfer-based attack, using a surrogate model to generate adversarial examples that can generalize to the target model. Specifically, let an available source VLP model be denoted as $F_S$, consisting of an image encoder $f_I$ and a text encoder $f_T$. Given an image-text pair $(x, t)$, the objective is to learn adversarial examples $x' = x + \delta$ and/or $t'$ that mislead the target model to wrongly align images and texts. Here $\delta$ is the pixel-level perturbation. To ensure imperceptibility, the magnitude of image perturbations

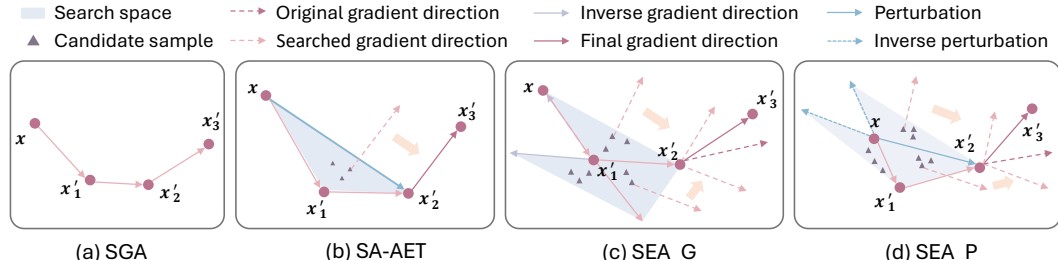

Figure 2: Comparison of (a) SGA (Lu et al., 2023), (b) SA-AET (Jia et al., 2025), (c) Gradient-based SEA, and (d) Perturbation-based SEA. SGA is constrained to the current adversarial trajectory, while SA-AET samples a new update direction from an intersection region constructed using the original, most recent, and current perturbations. In contrast, our methods broaden the adversarial search space to explore diverse update directions, effectively mitigating overfitting and avoiding local optima.

is constrained by the $l_\infty$-norm, i.e., $||\delta||_\infty \leq \epsilon_I$. Text perturbations are restricted by the maximum number of words that can be modified in a sentence, denoted as $\epsilon_T$.

## 2.2 MOTIVATION

The goal of a VLP attack is to disrupt cross-modal interactions, causing the target model to incorrectly associate images and texts. For image attacks, a common approach is to enlarge the embedding distance between matched image–text pairs:

$$\max_\delta \ell(f_I(x + \delta), f_T(t)), \ s.t., \ ||\delta||_\infty \leq \epsilon_I, \tag{1}$$

where $\ell$ represents a loss function, e.g., a cosine similarity-based loss.

The iterative gradient-based method is utilized to optimize the objective at the time step $n$:

$$\delta_{k+1} = \delta_k + \lambda g_k, \ g_k = \text{sign}\left(\frac{\nabla_{\delta_k}\ell(f_I(x + \delta_k), f_T(t))}{\|\nabla_{\delta_k}\ell(f_I(x + \delta_k), f_T(t))\|}\right). \tag{2}$$

According to the results in Table 1, even this simplest method is effective in the white-box setting, successfully attacking the target models used during training (i.e., the source model). However, its performance drops significantly when transferred to other models, indicating overfitting to the source model and limited generalization. In other words, following the vanilla optimization trajectories can lead to local optima. To address this, some approaches introduce strategies such as data augmentation (Lu et al., 2023; Yin et al., 2023), and adversarial example diversification (Gao et al., 2024), to regularize the gradient and explore alternative update directions. These techniques help reduce overfitting and improve black-box attack performance, but transferability remains constrained by the restricted search space. Specifically, input augmentation increases data diversity and acts as a form of model regularization, stabilizing update directions and mitigating overfitting. Despite its effectiveness, augmentation confines updates to a narrow subspace along the current adversarial trajectory, limiting exploration of diverse directions. Concurrent work (Jia et al., 2025; Gao et al., 2024) attempts to address this by introducing a new gradient direction learned from adversarial examples selected within an adversarial evolution region. However, these methods remain constrained in selection scope. They also rely on a single gradient for each adversarial update, failing to make use of diverse update directions. This limits the search space for transferable perturbations.

A similar phenomenon occurs in text attacks. Current methods typically rely on word substitution, where the most important words for a model's predictions are identified and replaced with other words, e.g., context-aware word substitutions (Lu et al., 2023; Yin et al., 2023; Jia et al., 2025). However, substituting only the most important word does not necessarily yield the most effective attack. Text representations are influenced by multiple factors, including the semantics of individual words, substitution candidates and their contextual relationships. Single-word substitution approaches consider only one word, overlooking other potentially influential replacements and

broader word interactions. Figure 1 demonstrates that focusing solely on the most important word is less effective than our method, which targets a less important word by considering more factors.

These limitations in both image and text modalities highlight a core challenge: existing methods explore only a restricted search space, which limits transferability. This motivates the need for a unified framework that systematically expands the search space to enable more transferable attacks.

## 2.3 IMAGE ATTACK

The proposed image attack aims to generate novel optimization directions using historical information, thereby avoiding following the original optimization trajectory that would lead to local optima. At each optimization step, gradients and perturbations for attacking the target model are computed. Instead of discarding this information after each update, SEA leverages it to expand the adversarial search space and reduce overfitting. Specifically, new search directions are explored from historical gradients and perturbations, with a selection strategy introduced to promote diverse yet effective updates. Within this framework, we propose two strategies: gradient-based expansion and perturbation-based expansion.

**Gradient-based search space expansion.** At optimization step $k$, we collect the set of past gradients along with the current gradient, denoted as $\{g(x + \delta_1), g(x + \delta_2), ..., g(x + \delta_{k-1})\}$. The adversarial search space is constructed, denoted as $\mathbb{G}_k = \{\hat{g}_k^1, \hat{g}_k^2, ..., \hat{g}_k^d\}$ with $d$ is the number of new gradients. $\hat{g}_k^d$ represents a candidate search direction derived from the combination of historical and current gradients:

$$\hat{g}_k = \sum_{i=1}^{k} \mu_i^{k,d} g(x + \delta_{i-1}), \ s.t. \sum_{i=1}^{k} \mu_i^{k,d} = 1. \tag{3}$$

**Perturbation-based search space expansion.** Similarly, we construct an adversarial search space based on past perturbations $\{\delta_1, \delta_2, ..., \delta_{k-1}\}$, denoted as $\Delta_k = \{\hat{\delta}_k^1, \hat{\delta}_k^2, ..., \hat{\delta}_k^d\}$, where

$$\hat{\delta}_k = \sum_{i=1}^{k} \eta_i^{k,d} \delta_{i-1}, \ s.t. \sum_{i=1}^{k} \eta_i^{k,d} = 1. \tag{4}$$

To encourage broader exploration and increase diversity in search directions, we employ two types of coefficient sets for both $\mu$ and $\eta$. The first is sampled from a uniform distribution $[0, 1)$, while the second is drawn from a standard normal distribution $\mathcal{N}(0, 1)$. In both cases, the coefficients are normalized so that their sum equals one.

**Cross-model interaction guided selection.** To ensure that novel update directions maintain attack effectiveness, we introduce a cross-model interaction guided selection strategy. Specifically, after gradient- or perturbation-based search space expansion, adversarial spaces are generated for each coefficient set, denoted as, i.e., $\mathbb{G}_k^1, \mathbb{G}_k^2, \Delta_k^1, \Delta_k^2$. The selection criterion is formulated as follows:

$$\mathbb{G}_k^{m+} = \{\hat{g}_k^m \in \mathbb{G}_k^m \mid \ell(f_I(x + \delta_k^m), f_T(t)) > \ell(f_I(x), f_T(t))\}, \ s.t. \ \delta_k^m = \lambda \hat{g}_k + \delta_{k-1}, \ m \in \{1, 2\},$$
$$\Delta_k^{m+} = \{\hat{\delta}_k^m \in \Delta_k^m \mid \ell(f_I(x + \hat{\delta}_k^m), f_T(t)) > \ell(f_I(x), f_T(t))\}, \ s.t. \ m \in \{1, 2\},$$
$$\tag{5}$$

where $\lambda$ is the step size. The perturbation generated by the gradient-based expansion method is $\hat{\delta}_k^m = \lambda \hat{g}_k^m + \delta_{k-1}$.

**Perturbation Learning.** At each step $k$, for each search space expansion scheme, we obtain three types of adversarial perturbations: the original perturbation and two newly explored ones, denoted as $\tilde{\Delta}_k$. These perturbations form an expanded attack space, which is leveraged to learn new perturbations. The overall objective for perturbation learning is formulated as follows:

$$\delta_{k+1} = \delta_k + \sum_{\tilde{\delta}_k \in \tilde{\Delta}_t} \epsilon_{k+1}^{(\tilde{\delta}_k)},$$
$$s.t. \ \epsilon_{k+1}^{(\tilde{\delta}_k)} = \tilde{\delta}_{k+1} - \tilde{\delta}_k, \ \tilde{\delta}_{k+1} = \arg\max_{\tilde{\delta}_k} \ell(f_I(x + \tilde{\delta}_k), f_T(t)). \tag{6}$$

### 2.3.1 Theoretical Analysis

Adversarial transferability is influenced by multiple factors. Our method seeks to explore a broader space of update directions by leveraging both current and historical gradient or perturbation information, thereby reducing the risk of overfitting and becoming trapped in local optima. A selection mechanism is incorporated to ensure that the generated updates remain effective. Beyond this empirical motivation from optimization dynamics, we provide theoretical analysis showing that our approach reduces interactions between perturbation units compared to existing methods, including (Lu et al., 2023; Zhang et al., 2024; Yin et al., 2023), further enhancing transferability.

Wang et al. (2020) demonstrated that interactions between perturbation units are negatively correlated with adversarial transferability. A common approach to quantify such interactions is via a Shapley value–based interaction index. Specifically, let $\mathcal{D}$ denote the set of all units (or dimensions) within a perturbation, and let $S \subseteq \mathcal{D}$ be a subset. Define $S_{ij} = \{i, j\}$ as a combined unit of $i$ and $j$. The set $\mathcal{D}' = \mathcal{D} \setminus \{i, j\} \cup S_{ij}$ represents the perturbation units in which $i$ and $j$ are treated jointly as a combined unit $S_{ij}$. Let $\phi(\cdot)$ denote the Shapley value. The interaction between perturbation units $i$ and $j$ is then defined as:

$$I_{ij}(\delta) = \phi(S_{ij}|\mathcal{D}') - [\phi(i|\mathcal{D} \setminus \{j\}) + \phi(j|\mathcal{D} \setminus \{i\})], \tag{7}$$

where $\phi(S_{ij}|\mathcal{D}')$ represents the contribution of $S_{ij}$. $\phi(i|\mathcal{D} \setminus \{j\})$ and $\phi(j|\mathcal{D} \setminus \{i\})$ denotes the contributions of units $i$ and $j$ when the other is absent, respectively. The interaction strength is indicated as $|I_{ij}(\delta)|$. A positive $I_{ij}(\delta)$ indicates that $\delta_i$ and $\delta_j$ interact positively, whereas $I_{ij}(\delta) < 0$ implies that their co-occurrence has a negative influence.

Based on this definition of interactions, we present the following key propositions:

**Proposition 1.** *Adversarial perturbations generated by SEA are defined as $\delta_k = \alpha \sum_{i=1}^{k} g_i$, where*

$$g_k = \begin{cases} \sum_{\tilde{\delta} \in \tilde{\Delta}_k} \nabla_\delta \ell(f_I(x + \tilde{\delta}), f_T(t)), & \text{if } k > 1, \\ g(x), & \text{if } k = 1. \end{cases} \tag{8}$$

*The adversarial perturbation generated by a general multi-step attack (e.g., SGA) is $\delta_k^{msa} = \alpha \sum_{i=0}^{k-1} \nabla_x \ell(f_I(x + \delta_i^{msa}), f_T(t))$. The interaction of perturbations generated by the proposed method is smaller than that of the general multi-step attack, i.e. $\mathbb{E}_{a,b}[I_{ab}(\delta_k)] \leq \mathbb{E}_{a,b}[I_{ab}(\delta_k^{msa})]$. (Detained proof can be found in Appendix B.)*

In Section C.1, we further compare the interaction values of perturbations generated by different attacks, demonstrating the effectiveness of the proposed method.

### 2.4 Text Attack

Instead of limiting substitutions to only the most important word (Lu et al., 2023; Yin et al., 2023; Gao et al., 2024; Zhang et al., 2025), the proposed method considers multiple important words and evaluates the effect of their replacements. This can balance the efficiency and effectiveness. Specifically, the method first identifies the top-$k$ important words and generates candidate substitutions for each using a pre-trained BERT model (Li et al., 2020). To ensure imperceptibility of these substitutions, we select candidate words whose similarity to the original words exceeds a threshold, e.g., $\text{sim}(w', w) > \tau$. Let the resulting set of modified texts be $\tilde{T} = \cup_{j=1}^{p}\{\tilde{t}_j^1, \tilde{t}_j^2, ..., \tilde{t}_j^l\}$, where $l$ is the number of candidate replacements per word and $p$ is the number of selected important words. Among these candidates, the final substitution is chosen to maximize the feature distance between the modified text and its paired images:

$$t' = \arg\max_{\tilde{t} \in \tilde{T}} \ell(f_I(x), f_T(\tilde{t})), \tag{9}$$

By expanding the scope of word substitutions, the proposed SEA leverages the semantics of individual words and candidate substitutions. At the same time, it also implicitly accounts for interactions between words to effectively modify the original text's meaning. In our experiments, we demonstrate that SEA outperforms existing methods in terms of attack effectiveness.

See Algorithm 1 in the Appendix for the complete procedure.

# 3 EXPERIMENTS

## 3.1 SETTINGS

**Datasets, Tasks, and Models.** We comprehensively evaluate SEA and baseline approaches on diverse datasets, including Flickr30K (Plummer et al., 2015), MSCOCO (Lin et al., 2014), and RefCOCO+ (Yu et al., 2016). The evaluation covers multiple tasks, including image–text retrieval, image captioning, and visual grounding. Four widely used VLP models, namely CLIP (Radford et al., 2021), ALBEF (Li et al., 2021), TCL (Yang et al., 2022), and BLIP (Li et al., 2022), are employed as source and target models. For CLIP, both ViT-B/16 (Dosovitskiy et al., 2020) and ResNet101 (He et al., 2016) backbones are considered. For fair comparison, configurations follow recent studies (Zhang et al., 2022; Lu et al., 2023; Yin et al., 2023; Zhang et al., 2024).

**Implementation Details.** The perturbation magnitudes are set uniformly to $\epsilon_I = 2/255$ for images and $\epsilon_T = 1$ for text. For image attacks, we employ Projected Gradient Descent (PGD) (Madry et al., 2018) with 10 iterations, a step size of $0.5/2.25$, and a batch size of 4. We set the number of update directions in each set and important words to $d = 5, p = 5$. Following the original settings, we set $\tau = 0.3$. To ensure reproducibility, all experiments are conducted with five fixed seeds $[42 - 46]$, and the results are reported as averages across these runs.

**Baselines.** We compare the proposed method with state-of-the-art approaches, including Co-Attack (Zhang et al., 2022), SGA (Lu et al., 2023), VLATTACK (Yin et al., 2023), DRA (Gao et al., 2024), and SA-AET (Jia et al., 2025).

**Evaluation Metric.** The attack success rate (ASR) is employed to evaluate all methods on image-text retrieval, quantifying the proportion of adversarial examples that successfully induce incorrect predictions. For other tasks, the effect of attacks is assessed by measuring the performance drop of the target models before and after the attack.

## 3.2 EXPERIMENTAL RESULTS

### 3.2.1 CROSS-MODEL TRANSFERABILITY

We evaluate the adversarial transferability of attacks across different models in image-text retrieval, which identifies relevant items for a given query by ranking their distances in the embedding space. In experiments, we consider different source and target models, including CLIP$_{\text{ViT-B/16}}$, CLIP$_{\text{ResNet101}}$, ALBEF, and TCL, with results summarized in Table 1. In the table, TR denotes image-to-text retrieval, while IR denotes text-to-image retrieval. Due to space constraints, only R@1 results are reported. Several key observations emerge from the results. We evaluate the proposed method in two variants: SEA-G (gradient-based expansion), SEA-P (perturbation-based expansion).

First, the compared methods exhibit limited adversarial transferability. While they achieve promising results on the source model, their performance on target models is consistently inferior. In contrast, the proposed method achieves the best performance across all cases, demonstrating that expanding the search space reduces overfitting and improves transferability. Second, adversarial transferability varies across models due to differences in architecture, training objectives, and learned representations. As shown in Table 1, transfers between models with similar architectures and objectives, such as from CLIP$_{\text{ViT-B/16}}$ to CLIP$_{\text{ResNet101}}$, are more effective than transfers between less similar models, e.g., from CLIP$_{\text{ViT-B/16}}$ to ALBEF. This supports the observation that model-specific features limit transferability. Third, transferability is not symmetric: an attack that transfers effectively from one model to another (e.g., from ALBEF to CLIP) does not necessarily transfer in the reverse direction. This asymmetry is particularly interesting and suggests a promising direction for future research.

### 3.2.2 CROSS-TASK TRANSFERABILITY

To further demonstrate the superiority of the proposed method across different tasks, we leverage adversarial examples learned from image-text retrieval to attack models for visual grounding and image captioning. Compared to image-text retrieval, these tasks consider more fine-grained relationships between images and texts: visual grounding aims to match text to corresponding regions or objects in an image, whereas image captioning generates detailed descriptions of image content,

Table 1: Attack success rate (%) regarding the average of R@1 in image-text retrieval on Flickr30K. * highlights white-box attack results, and **bold** indicates the best performance.

| Source Model | Method | CLIP$_{ViT-B/16}$ TR R@1 | CLIP$_{ViT-B/16}$ IR R@1 | CLIP$_{ResNet101}$ TR R@1 | CLIP$_{ResNet101}$ IR R@1 | ALBEF TR R@1 | ALBEF IR R@1 | TCL TR R@1 | TCL IR R@1 |
|---|---|---|---|---|---|---|---|---|---|
| CLIP$_{ViT-B/16}$ | Co-Attack | 94.64* | 96.09* | 28.59 | 40.34 | 9.91 | 24.16 | 11.17 | 25.55 |
|  | SGA | **99.14*** | **99.19*** | 41.00 | 48.13 | 12.41 | 27.69 | 16.42 | 30.74 |
|  | VLATTACK | 98.16* | 98.87* | 36.14 | 43.36 | 10.01 | 26.03 | 13.07 | 27.83 |
|  | DRA | 98.53* | 98.71* | 44.32 | 49.91 | 12.72 | 29.42 | 13.81 | 30.69 |
|  | SA-AET | 98.90* | 99.03* | 44.70 | 50.66 | 14.29 | 30.52 | 15.49 | 30.69 |
|  | SEA-P | 99.26* | 99.07* | 53.43 | **58.50** | **17.62** | **33.07** | 18.86 | 34.69 |
|  | SEA-G | 98.40* | 98.23* | **53.51** | 58.49 | 17.10 | 31.95 | **18.97** | **34.71** |
| CLIP$_{ResNet101}$ | Co-Attack | 28.59 | 40.34 | 98.08* | 98.52* | 9.38 | 23.90 | 11.70 | 25.12 |
|  | SGA | 32.52 | 44.30 | 97.80* | 97.84* | 11.57 | 25.28 | 13.49 | 27.83 |
|  | VLATTACK | 32.88 | 43.23 | 94.13* | 96.78* | 3.86 | 9.33 | 8.54 | 13.71 |
|  | DRA | 33.87 | 46.01 | 98.72* | 98.77* | 12.41 | 26.59 | 13.80 | 29.07 |
|  | SA-AET | 35.09 | 46.49 | 99.62* | 99.55* | 12.13 | 27.90 | 13.49 | 28.81 |
|  | SEA-P | 41.35 | **53.54** | 99.87* | 99.73* | **14.29** | **29.96** | 16.65 | **32.57** |
|  | SEA-G | **41.60** | 53.45 | 99.35* | 99.83* | 13.45 | 29.84 | **16.97** | 32.33 |
| ALBEF | Co-Attack | 23.98 | 35.50 | 25.72 | 38.65 | 80.24* | 87.68* | 15.13 | 29.02 |
|  | SGA | 33.43 | 44.49 | 35.15 | 48.32 | 97.72* | 97.52* | 46.01 | 55.19 |
|  | VLATTACK | 36.56 | 45.78 | 35.63 | 47.17 | 94.13* | 96.78* | 41.62 | 52.81 |
|  | DRA | 38.93 | 48.90 | 42.53 | 50.37 | 96.98* | 96.77* | 50.79 | 61.33 |
|  | SA-AET | 40.12 | 50.48 | 44.19 | 52.10 | 97.39* | **98.04*** | 57.06 | 64.07 |
|  | SEA-P | **42.33** | 52.90 | 46.74 | **55.09** | 96.66* | 97.59* | 59.43 | 66.69 |
|  | SEA-G | 41.35 | **53.16** | **47.51** | 54.89 | 95.41* | 96.26* | **60.48** | **67.50** |
| TCL | Co-Attack | 28.15 | 41.66 | 30.23 | 44.46 | 22.89 | 20.90 | 79.24* | 85.62* |
|  | SGA | 34.39 | 44.62 | 37.53 | 48.94 | 48.82 | 60.18 | 98.33* | 99.00* |
|  | VLATTACK | 36.44 | 21.20 | 36.91 | 47.86 | 34.20 | 46.80 | 94.13* | 96.78* |
|  | DRA | 37.50 | 50.58 | 43.81 | 51.17 | 52.14 | 63.47 | 98.31* | 98.88* |
|  | SA-AET | 38.53 | 51.64 | 45.87 | 52.72 | 56.35 | 68.18 | 98.31* | 98.76* |
|  | SEA-P | 42.09 | 53.77 | 47.13 | 55.44 | 60.27 | 71.21 | **98.42*** | **99.19*** |
|  | SEA-G | **42.21** | **54.16** | **48.02** | **55.68** | **61.94** | **72.33** | 98.21* | 98.52* |

Table 2: Performance on visual grounding under different attacks on RefCOCO+. ALBEF for image-text retrieval serve as the source model, and ALBEF for isual grounding serves as the target model. "Baseline" refers to the target model's performance on clean data. Smaller values indicate better adversarial transferability, and **bold** highlights the best results.

|  | Baseline | Co-Attack | SGA | VLATTACK | DRA | SA-AET | SEA-P | SEA-G |
|---|---|---|---|---|---|---|---|---|
| Val | 58.46 | 54.26 | 50.56 | 56.67 | 49.32 | 47.44 | **45.51** | 46.43 |
| TestA | 65.89 | 61.80 | 57.42 | 65.37 | 56.48 | 53.27 | **51.62** | 51.90 |
| TestB | 46.25 | 43.81 | 36.61 | 45.28 | 40.15 | 38.58 | **35.84** | 37.29 |

including objects, background, and their relationships. Tables 2 and 3 report the results, showing that the proposed method more effectively degrades target model performance. This demonstrates that expanding the search space mitigates overfitting to model-specific features and improves transferability across tasks and models.

## 3.3 ABLATION STUDY

### 3.3.1 IMPACT OF CANDIDATE SELECTION NUMBER

The proposed method leverages diverse update information in the image modality and explores a broader range of potentially influential words. Taking SEA-G as an example, we evaluate performance and computational cost when varying the number of new gradients ($d$) and the number of word replacements ($p$), using a workstation equipped with an NVIDIA GTX 1080 Ti GPU. The results are reported in Figure 4. From the Figure, it can be observed that expanding the search space generally improves performance while maintaining efficiency. For the image modality, increasing $d$ yields clear gains, though larger values introduce additional computational overhead, with $d = 5$ offering a good trade-off between effectiveness and efficiency. In the text modality, increasing $p$ enhances performance, but when $p$ exceeds 5, the gains become less pronounced. These findings verify the effectiveness of the proposed method in leveraging the top-$p$ important words.

Table 3: Performance in the image captioning task under various attacks on MSCOCO. ALBEF for image-text retrieval serves as the source model, while BLIP built for image captioning is used as the target model. "Baseline" refers to the target model's performance on clean data. Smaller values indicate better adversarial transferability, and **bold** highlights the best results.

| | Baseline | Co-Attack | SGA | VLATTACK | DRA | SA-AET | SEA-P | SEA-G |
|---|---|---|---|---|---|---|---|---|
| B@4 | 39.59 | 37.27 | 34.23 | 36.23 | 33.62 | 33.37 | **32.16** | 32.55 |
| METEOR | 30.87 | 29.67 | 28.41 | 28.18 | 27.85 | 27.60 | **26.92** | 27.19 |
| ROUGE_L | 59.67 | 57.98 | 56.16 | 57.46 | 55.62 | 54.69 | **54.38** | 54.72 |
| CIDEr | 133.02 | 125.19 | 115.45 | 126.31 | 111.30 | 107.65 | **105.17** | 106.88 |
| SPICE | 23.51 | 22.58 | 21.46 | 22.49 | 20.87 | 20.52 | **19.92** | 20.25 |

Table 4: Effects of selection number. The attack success rate of R@1 on image-text retrieval is reported. CLIP$_{\text{ViT-B/16}}$ is adopted as the source model. * highlights white-box attack results.

| Target Model | CLIP$_{\text{ViT-B/16}}$ | | CLIP$_{\text{ResNet101}}$ | | ALBEF | | TCL | | Computation time per sample |
|---|---|---|---|---|---|---|---|---|---|
| Parameter | TR R@1 | IR R@1 | TR R@1 | IR R@1 | TR R@1 | IR R@1 | TR R@1 | IR R@1 | |
| $d$=1 | 96.69* | 99.71 | 49.36 | 51.95 | 15.89 | 31.32 | 17.25 | 32.59 | 1.084s |
| $d$=3 | 99.08* | 99.11* | 52.60 | 55.65 | 16.25 | 32.13 | 17.68 | 33.15 | 2.177s |
| $d$=5 | 98.40* | 98.23* | 53.51 | 58.49 | 17.10 | 31.95 | 18.97 | 34.71 | 3.070s |
| $d$=7 | 98.70* | 99.26* | 55.54 | 59.94 | 18.03 | 33.33 | 20.09 | 35.41 | 4.843s |
| $d$=9 | 99.02* | 99.67* | 56.00 | 59.81 | 18.29 | 33.40 | 21.26 | 35.36 | 6.511s |
| $p$=1 | 99.51* | 99.71* | 47.36 | 50.95 | 15.37 | 31.22 | 17.70 | 31.88 | 0.188s |
| $p$=3 | 97.91* | 97.97* | 45.59 | 52.01 | 15.22 | 30.71 | 16.33 | 32.74 | 0.190s |
| $p$=5 | 98.40* | 98.23* | 53.51 | 58.49 | 17.10 | 31.95 | 18.97 | 34.71 | 0.199s |
| $p$=7 | 97.89* | 98.61* | 53.47 | 58.40 | 17.18 | 31.79 | 18.84 | 34.79 | 0.212s |
| $p$=9 | 98.68* | 99.01* | 53.85 | 58.57 | 17.39 | 31.99 | 18.88 | 35.00 | 0.225s |

Table 5: Ablation study of different components on Flickr30K for image-text retrieval. The attack success rate of R@1 on image-text retrieval is reported. CLIP$_{\text{ViT-B/16}}$ is adopted as the source model. * highlights white-box attack results.

| Target Model | CLIP$_{\text{ViT-B/16}}$ | | CLIP$_{\text{ResNet101}}$ | | ALBEF | | TCL | |
|---|---|---|---|---|---|---|---|---|
| Method | TR R@1 | IR R@1 | TR R@1 | IR R@1 | TR R@1 | IR R@1 | TR R@1 | IR R@1 |
| PGD | 69.82* | 74.87* | 5.11 | 7.82 | 2.40 | 5.03 | 4.32 | 7.14 |
| SEA-P-Img | 95.21* | 95.94* | 17.75 | 20.33 | 6.57 | 10.92 | 7.38 | 13.52 |
| SEA-G-Img | 97.55* | 98.03* | 18.02 | 20.65 | 6.15 | 10.34 | 7.06 | 13.24 |
| BERT-Attack | 17.11* | 39.05* | 30.27 | 37.39 | 9.18 | 22.64 | 9.59 | 24.05 |
| SEA-Txt | 45.40* | 53.35* | 40.74 | 45.11 | 11.78 | 26.94 | 13.38 | 28.71 |
| SEA-P w/o TE | 98.90* | 98.81* | 45.85 | 53.76 | 15.02 | 31.24 | 16.33 | 33.45 |
| SEA-G w/o TE | 97.18* | 97.65* | 46.49 | 53.47 | 15.54 | 32.02 | 16.44 | 33.48 |
| SEA w/o IE | 99.51* | 99.71* | 47.36 | 50.95 | 15.37 | 31.22 | 17.70 | 31.88 |
| SEA-P | 99.26* | 99.07* | 53.43 | 58.50 | 17.62 | 33.07 | 18.86 | 34.69 |
| SEA-G | 98.40* | 98.23* | 53.51 | 58.49 | 17.10 | 31.95 | 18.97 | 34.71 |
| SEA w/ Momentum | 99.26* | 99.19* | 55.56 | 60.62 | 17.83 | 33.49 | 20.65 | 35.86 |
| SEA w/ P + G | 99.39* | 99.10* | 54.36 | 62.02 | 18.98 | 34.03 | 20.55 | 36.19 |

### 3.3.2 MODULE ABLATION

To evaluate the contribution of each component, we conduct a comprehensive ablation study. For the image modality, we construct variants that impose only image attacks, denoted SEA-G-Img and SEA-P-Img for the gradient- and perturbation-based variants, respectively. For the text modality, we report the text-attack variant, SEA-Txt. We also test variants that remove the expansion component in the image and text modalities, denoted SEA-P w/o TE, SEA-G w/o TE, and SEA w/o IE. Additionally, we combine SEA with momentum to assess its adaptability. Table 5 summarizes the results of image-text retrieval on Flickr30K using CLIP$_{\text{ViT-B/16}}$ as the source model. The findings show consistent gains from each component, with SEA w/ P+G achieving the best performance, highlighting the effectiveness and complementarity of our expansions. Moreover, combining the proposed search-space expansion with momentum yields further improvements, supporting our motivation that exploring a broader adversarial space mitigates overfitting and enhances transferability.

## 4 RELATED WORK

### 4.1 VISION-LANGUAGE PRE-TRAINED MODELS

Real-world applications frequently involve heterogeneous data modalities, with vision-language tasks such as image–text retrieval, image captioning, and visual question answering becoming increasingly prevalent (Zhang et al., 2021a; Wang et al., 2017; 2025). These tasks demand models that are not only powerful but also generalizable, capable of understanding both visual and textual information as well as their complex interactions. However, training task-specific models from scratch each time is often impractical due to prohibitive computational costs and the scarcity of annotated data. Vision-language pre-training (VLP) has emerged as a key paradigm to address this need, endowing models with transferable multimodal representations through large-scale unsupervised pretraining (Radford et al., 2021; Li et al., 2022; 2021; Yang et al., 2022). For example, CLIP (Radford et al., 2021) projects different modalities into a unified feature space using unimodal encoders. BLIP (Li et al., 2022) refines the dataset by removing noisy captions to enhance performance. ALBEF (Li et al., 2021) and TCL (Yang et al., 2022) introduce an additional multimodal encoder, aiming to learn universal representations across modalities for better alignment.

### 4.2 ADVERSARIAL ATTACK

With the increasing deployment of VLP models, ensuring their reliability is essential, as real-world applications inevitably face uncertainty and security threats. Among these, adversarial attacks are a pressing concern, which aims to craft imperceptible perturbations that mislead models (Szegedy et al., 2014; Madry et al., 2018; Moosavi-Dezfooli et al., 2017). Evaluating robustness under such attacks is thus critical. More importantly, adversarial attacks provide valuable insights into the inner mechanisms of deep models, facilitating the design of systems that are both accurate and resilient.

In practical scenarios, the target model may not always be accessible, i.e., black-box cases, and learning adversarial examples for each new model is often impractical. To address this, existing approaches typically rely on transfer-based methods, crafting adversarial examples on a source model that generalize to unseen targets. However, developing transferable attacks for VLP models remains difficult due to variations in architecture, objectives, and training strategies across tasks. Several approaches have been proposed to address these challenges. For instance, in the image modality, (Lu et al., 2023; Zhang et al., 2024; Gao et al., 2024) leverage data augmentation and exploit cross-modal interactions, while (Gao et al., 2024) further promote the diversity of adversarial examples. (Yin et al., 2023) exploit block-wise gradients to mislead fine-grained features. Despite these advances, existing strategies remain limited. Data augmentation is still constrained to the current adversarial trajectory, failing to leverage more diverse update information that could enhance transferability. Block-wise attack introduces additional computational overhead and still focuses on the current optimization path, which limits its effectiveness. Ensemble-based methods require access to multiple models and incur substantial computational costs, making them impractical for real-world use. In text-based attacks, existing methods typically identify the most important words for a prediction and replace them with similar alternatives (Lu et al., 2023; Yin et al., 2023; Zhang et al., 2025). However, considering only individual word importance often fails to produce the most effective adversarial substitutions due to the diversity of candidates and complex word interactions. Expanding the search space is therefore necessary to identify more impactful replacements.

## 5 CONCLUSIONS

In this paper, we identify that the restricted adversarial search space is a key factor limiting transferability in vision–language pretraining (VLP) models. To address this, we propose a novel and effective attack, termed Search-space Expansion Attack (SEA), which expands adversarial search directions in both image and text modalities. For images, SEA leverages historical update information to explore more effective search directions, avoiding overfitting and local optima. For text, SEA goes beyond considering only the most salient word and extends the search across multiple potentially influential words, accounting for both individual importance and word interactions. Through extensive experiments and rigorous theoretical analysis, we demonstrate that SEA achieves superior adversarial transferability compared to current methods.

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

---

**Algorithm 1:** Search-space Expansion Attack

---

**Require**: Training data $\mathcal{D} = \{(x_i, t_i)\}_{i=1}^n$, mini-batch size $b$, iteration times $K$, parameters
$\epsilon_I, \epsilon_T, \tau, \lambda, p, d, l$;
**Require**: Randomly initialize $\delta_0, g_0$;
**Require**: A source model $F_s = \{F_I, F_T\}$;
// Training;
// Exploit diverse matching captions for each image to augment the dataset;
// **Stage 1: Generate adversarial texts**:
// Calculating the importance of each original word and accordingly ranking them;
$w' \leftarrow \text{sim}(w', w) > \tau$ // Identify top-$l$ vulnerable vocabulary that similar to the original word;
$\tilde{T} = \cup_{j=1}^p \{\tilde{t}_j^1, \tilde{t}_j^2, ..., \tilde{t}_j^l\}$ // Get candidate word set using BERT model;
$t' = \arg\max_{\tilde{t} \in \tilde{T}} \ell(f_I(x), f_T(\tilde{t}))$ // Select the substitution word that satisfies Eq.9 to obtain the
adversarial text;
// **Stage 2: Generate adversarial images**:
**for** $k = 1 \to K$ **do**
  // Initialize adversarial image
  $\hat{g}_k = \sum_{i=1}^k \mu_i^{k,d} g(x + \delta_{i-1})$, or $\hat{\delta}_k = \sum_{i=1}^k \eta_i^{k,d} \delta_{i-1}$ // Regularize the current
    gradient/perturbation and generate diverse update directions according to Eq.3 and 4;
  $\mathbb{G}_k^{m+} = \{\hat{g}_k^m \in \mathbb{G}_k^m \mid \ell(f_I(x + \delta_k^m), f_T(t)) > \ell(f_I(x), f_T(t))\}$, $s.t.$ $\delta_k^m =$
    $\lambda\hat{g}_k + \delta_{k-1}$, $m \in \{1, 2\}$, or
  $\Delta_k^{m+} = \{\hat{\delta}_k^m \in \Delta_k^m \mid \ell(f_I(x + \hat{\delta}_k^m), f_T(t)) > \ell(f_I(x), f_T(t))\}$, $s.t.$ $m \in \{1, 2\}$, // Select
    the most effective combinations according to Eq.5;
  $\tilde{\Delta}_k \to \mathbb{G}_k^{m+}, \Delta_k^{m+}, \delta_k$ // Construct the final perturbation set;
  $\delta_{k+1} = \delta_k + \sum_{\tilde{\delta}_k \in \tilde{\Delta}_t} \epsilon_{k+1}^{(\tilde{\delta}_k)}$, $s.t.$ $\epsilon_{k+1}^{(\tilde{\delta}_k)} = \tilde{\delta}_{k+1} - \tilde{\delta}_k$, $\tilde{\delta}_{k+1} = \arg\max_{\tilde{\delta}_k} \ell(f_I(x + \tilde{\delta}_k), f_T(t))$ //
    Update new perturbations according to Eq.6;
  // Update adversarial image $x'_{k+1}$ with the generated perturbation $\delta_{k+1}$;
**end**
return $x', t'$;

---

# A  ALGORITHM

The overall procedure is summarized in Algorithm 1. In the first stage, the method identifies the top-$l$ keywords and searches for candidate substitutions that maximize the loss. In the second stage, adversarial images are generated. Specifically, the algorithm computes the gradient or perturbation required to update the adversarial examples, and then combines it with past gradients/perturbations to construct a diverse set of candidates. Finally, a selection strategy is applied to choose the most effective candidates for updating the perturbations. The process terminates upon reaching the maximum number of iterations.

# B  PROOF OF PROPOSITION 1

To simplify the problem setting, we omit certain techniques commonly used in adversarial attacks, such as gradient normalization and the clipping operation. For theoretical tractability, we assume that the combination weights of past gradients remain fixed across all optimization steps, allowing us to derive closed-form expressions for gradient and perturbation updates. To facilitate the proof, we further simplify the optimization process by considering a single combination of gradients and perturbations. Specifically, we study the proposed method under uniform gradient combination weights, i.e., $\mu_i^k = \frac{1}{k}$, which admits closed-form expressions for perturbation accumulation and interaction terms. Importantly, empirical results (as shown in Table C.1) indicate that the method consistently reduces perturbation interactions even under general, non-uniform weights, validating its broader effectiveness. We take perturbation-based SEA as an example. Specifically, during the perturbation search space expansion, the updated gradient of the proposed method is:

$$g_k \stackrel{\text{def}}{=} g(x + \frac{1}{k} \sum_{i=1}^k \delta_{i-1}) \tag{10}$$

where $t$ denotes the step. To simplify the notation, we use $g(x)$ to denote $\nabla_\delta \ell(f_I(x+\delta), f_T(t))$, i.e. $g(x) \overset{\text{def}}{=} \nabla_\delta \ell(f_I(x+\delta), f_T(t))$. The update of the perturbation in our attack at step $k$ is defined as:

$$\delta_k = \lambda \sum_{i=1}^{k} g_i, \tag{11}$$

**Proposition 2.** *Applying the Taylor expansion, we can have:*

$$g(x + \delta_{i-1}) = g(x) + H(x)\delta_{i-1} + r_{i-1}^1, \tag{12}$$

*The adversarial perturbation generated by our method via gradient descent can then be written as:*

$$g_k \overset{\text{def}}{=} a_k g + \lambda b_k H g + \tilde{R}_k^1,$$
$$\delta_k \overset{\text{def}}{=} c_k \lambda g + d_k \lambda^2 H g + \tilde{R}_k^1, \tag{13}$$

*where*

$$a_k = 1,$$
$$b_k = \frac{k-1}{2},$$
$$c_k = k,$$
$$d_k = \frac{(k-1)k}{4}, \tag{14}$$

*where $g$ denotes the gradient $g(x)$ and $H$ denotes the second-order Hessian matrix $H(x)$.*

*Proof.* $g_{k+1}$ is defined as the average of all past gradients, and $\delta_{k+1}$ is the corresponding combination of these gradients. Formally, we have:

$$g_k = g(x + \frac{1}{k}\sum_{i=1}^{k} \delta_{i-1})$$
$$= (g + \frac{1}{k}\sum_{i=1}^{k} H(c_{i-1}\lambda g + d_{i-1}\lambda^2 H g + \tilde{R}_{i-1}^1) + r_{i-1}^1) \tag{15}$$
$$= g + \lambda\frac{1}{k}(\sum_{i=1}^{k} c_{i-1})H g + \tilde{R}_t^2,$$

$$\delta_k = \lambda\sum_{i=1}^{k} g_i$$
$$= \lambda\sum_{i=1}^{k}(a_i g + \lambda b_i H g + \tilde{R}_i^1) \tag{16}$$

These expressions allow us to explicitly analyze the accumulation of gradient contributions and perturbation interactions, which leads to:

$$a_k = 1. \tag{17}$$

$$c_k = \sum_{j=1}^{k} a_j = k. \tag{18}$$

The coefficient of $Hg$, i.e., $b_k$ can be obtain by:

$$b_k = \sum_{i=1}^{k} \mu_i^k c_{i-1}. \tag{19}$$

$$d_k = \sum_{i=1}^{k} b_i$$
$$= \sum_{i=1}^{k} \sum_{j=1}^{i} \mu_j^k c_{j-1}. \tag{20}$$

At the same time, we have:

$$\delta_{k+1} = \delta_k + \lambda g_{k+1}$$
$$= c_k \lambda g + d_k \lambda^2 Hg + \tilde{R}_t^1 + \lambda(a_{k+1}g + \lambda b_{k+1} Hg + \tilde{R}_{k+1}^1) \tag{21}$$
$$= (c_k + a_{k+1})\lambda g + (d_t + b_{k+1})\lambda^2 Hg + \tilde{R}_{k+1}^2.$$

Thus, the coefficients update recursively as:

$$c_k + a_{k+1} = k + 1$$
$$= b_{k+1}. \tag{22}$$

$$d_t + b_{k+1} = \frac{k(k-1)}{4} + \frac{k}{2}$$
$$= \frac{k(k+1)}{4} \tag{23}$$
$$= d_{k+1}.$$

According to Section 2.3.1, the Shapley interaction between perturbation units $a, b$ in $\delta^m$ is calculated as follows:

$$I_{ab}(\delta_k) = \delta_{k,a}, H_{ab}\delta_{k,b}^{\text{sea}} + \tilde{R}_2(\delta^k),$$
$$= (c_k\lambda g_a + d_k\lambda^2 g^T H_{*a})H_{ab}(c_k\lambda g_b + d_k\lambda^2 g^T H_{*b}),$$
$$= c_k^2 \lambda^2 g_a g_b H_{ab} + c_k d_k \lambda^3 g_b H_{ab} g^T H_{*a} + c_k d_k \lambda^3 g_a H_{ab} g^T H_{*b} + O(H^2), \tag{24}$$
$$= k^2 \lambda^2 g_a g_b H_{ab} + \frac{k^2(k-1)}{4}\lambda^3 g_b H_{ab} g^T H_{*a} + \frac{k^2(k-1)}{4}\lambda^3 g_a H_{ab} g^T H_{*b} + O(H^2),$$

Here, $O(H^2)$ denotes second-order small terms with respect to $H$, which are neglected in the following calculations. According to (Wang et al., 2020), we have $\mathbb{E}_{a,b}[g_a g_b g^T H*b] \geq 0$. The interaction for perturbations learned by a general multi-step attack, e.g., SGA, is given by $I_{ab}(\delta_k^{\text{msa}}) = k^2\lambda^2 g_a g_b H_{ab} + \frac{k^2(k-1)}{2}\lambda^3 g_b H_{ab}g^T H_{*a} + \frac{k^2(k-1)}{2}\lambda^3 g_a H_{ab}g^T H_{*b} + O(H^2)$. As a result, $\mathbb{E}_{a,b}(I_{ab}(\delta_k) - I_{ab}(\delta_k^{\text{msa}})) = \mathbb{E}_{a,b}(-\frac{k^2(k-1)}{4}\lambda^3 g_b H_{ab}g^T H_{*a} - \frac{k^2(k-1)}{4}\lambda^3 g_a H_{ab}g^T H_{*b}) < 0$. Therefore, compared to the general multi-step attack, the proposed method induces fewer interactions, thereby promoting transferability.

$\square$

## C    MORE RESULTS

### C.1    INTERACTION COMPARISON

Table C.1 shows interaction values of perturbations by different attacks with CLIP$_{\text{ViT-B/16}}$ as the source model on the Flickr30K dataset and image-text retrieval. Results show that the proposed

Table 6: Average interaction of adversarial perturbations generated by different attacks with $CLIP_{ViT-B/16}$ as the source model on the Flickr30K dataset and image-text retrieval.

|  | Co-Attack | SGA | VLATTACK | DRA | SA-AET | SEA-P | SEA-G |
|---|---|---|---|---|---|---|---|
| Interaction | 0.054 | - 0.108 | - 0.99 | - 0.097 | - 0.149 | - 0.093 | - 0.087 |

method induces antagonistic interactions, with lower interaction strength compared to other methods, indicating fewer interactions between perturbation units. Combined with the superior adversarial transferability reported earlier, these findings highlight the method's advantage over existing approaches. It is also noted that adversarial transferability is influenced by multiple factors beyond interactions alone. For example, SA-AET achieves higher transferability than VLATTACK despite exhibiting stronger interactions. This is because VLATTACK considers block-wise similarity, which reduces interactions, whereas SA-AET increases the diversity of adversarial examples, preventing overfitting and thereby improving transferability.

### C.2 VISUALIZATION

In Figure 3, we provide several Grad-CAM (Selvaraju et al., 2017) visualization examples, where $CLIP_{ViT-B/16}$ and ALBEF are the source and target models. From the figure, it can be observed that the proposed method effectively shifts the model's attention through image and/or text attacks. Furthermore, the image perturbations remain imperceptible to human observers. In contrast, adversarial attacks in text are more noticeable due to the discrete nature of language tokens. Enhancing the invisibility of text attacks remains a promising direction for future research.

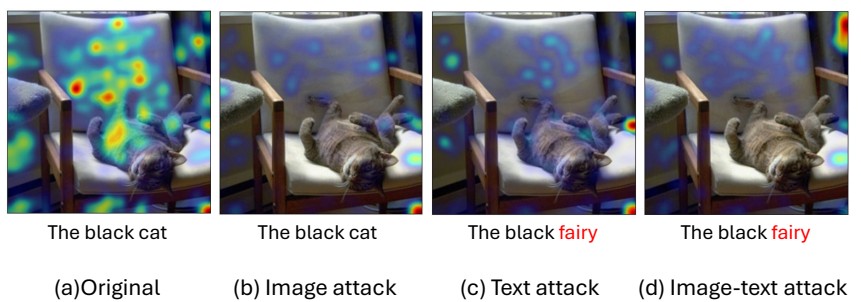

|  The black cat | The black cat | The black fairy | The black fairy |
|---|---|---|---|
| (a)Original | (b) Image attack | (c) Text attack | (d) Image-text attack |

Figure 3: A Grad-CAM visualization of adversarial examples generated by the proposed method, where $CLIP_{ViT-B/16}$ and ALBEF are the source and target models, where it can be seen that the proposed method can significantly shift the attention of target models.

## D LLM USAGE

LLMs were only used for minor language polishing.

