# OpenReview forum: "Enhancing Adversarial Transferability in Vision-Language Models via Search-Space Expansion"
_ICLR.cc/2026/Conference — Submitted to ICLR 2026_

### Official Review · Reviewer_b2fs · 2025-10-30

**Soundness:** 3
**Presentation:** 3
**Contribution:** 3
**Rating:** 2
**Confidence:** 4

**Summary:**

This paper focuses on the limited adversarial transferability of Vision-Language Pre-trained (VLP) models, attributing it to narrow adversarial search spaces that cause overfitting to source models. It proposes SEA, a framework that expands the search space for both modalities.

**Strengths:**

- Well written.
- SEA addresses transferability issues by expanding the search space for image and text modalities.
- SEA has good cross-task/model generalization ability.

**Weaknesses:**

- SEA's image module combines "current gradient " and "historical information ", but the paper does not isolate the contribution of each component.
- This paper claims SEA avoids "local optima", but no visualization of optimization trajectories is provided. Without this, it is impossible to verify if SEA truly escapes local optima or just converges to different ones.
- This paper lacks visualization results.
- This paper lacks a framework of the method, making it difficult to understand the SEA intuitively.
- The text module claims to "account for word interactions", but no quantitative evidence supports this.
- What is the significance of Proposition 1? It seems that the author merely conducted some mathematical derivations and did not provide an analysis of Proposition 1.

**Questions:**

- Please see "Weaknesses".

---

> ### Author Response · Authors · 2025-11-28
> **Response to Reviewer b2fs  (1)**
>
> We sincerely thank the reviewer for the constructive comments. We address the concerns point-by-point below.
> 1. The contribution of each component
>
> The motivation of our method is to expand the adversarial search space to mitigate overfitting. To achieve this efficiency, we propose to leverage existing available information to explore more search directions, which motivates use to incorporate past update information.
> In Table 5, we should the results of do not use both historical and current information for search space expansion, i.e., SEA w/o IE, which show the effectiveness of the proposed method. This is also verified in Table 4, which shows that using more combinations of past and current updating information can further enhance the performance.
>
> To further verify the effectiveness of only using historical information or current information for space expansion, we conduct additional experiments, where results are shown in Table 1.
>
> Table 1. Attack success rate $(\%)$regarding the average of R@1 in image-text retrieval on Flickr30K. {*} highlights white-box attack results.
>
> |Target Model|CLIP_ViT-B/16.|CLIP_RN101.|ALBEF. |TCL. |
> | Source Model |I2T |T2I| I2T| T2I. | I2T| T2I.| I2T| T2I. |
> |:---------:|:-----:|:------:|:-----:|:------:|:-----:|:------:|:-----:|:------:|
> |SEA-P w/o current perturbations| 99.51*| 99.36*| 52.20| 57.10| 16.31| 31.12| 17.06 |32.10|
> |SEA-P w/o past perturbations| 99.54*| 99.71*| 47.36| 50.95| 15.37| 31.22| 17.70  |31.88|
> |SEA-P| 99.26*| 99.07*| 53.43| 58.50| 17.62| 33.07| 18.86  |34.69|
>
> 2. Visualization of Local optima
>
> In our experiments, like Table 1, 2, and 3, is can be seen that all methods perform well in the white-box settings, while facing perform degradation in the black-box settings, i.e., attacks on other models and tasks. This means that these methods drop into local optima and encounter overfitting. In comparison, the proposed method achieves better black-box attack performance on different models and tasks, which indicates that it can help escapes local optima instead of converges to different ones.
> Visualization of optimization trajectories is not applicable here because SEA operates in a high-dimensional hybrid space combining continuous image perturbations with discrete text substitutions, which cannot be meaningfully reduced to 2D/3D plots.
>
> 3. Visualization results
>
> In Figures 1–3, we already provide representative examples of the text and image attacks as well as a Grad-CAM visualization. To further improve clarity, we will include additional visualization results in the revised version. We also welcome any specific recommendations from the reviewer regarding additional visualizations they would find helpful.

---

> > ### Author Response · Authors · 2025-11-28
> > **Response to Reviewer b2fs (2)**
> >
> > 4. A framework of the method
> >
> > We thank the reviewer for this comment. Algorithm 1 provides a step-by-step description of SEA, and Figures 1 and 2 visualize the proposed text and image attacks. To further improve clarity and provide an intuitive high-level understanding, we will add a dedicated framework diagram in the revised version that illustrates the overall workflow of SEA and how its components interact.
> >
> > 5. Word interactions
> >
> > We thank the reviewer for the comment. Unlike prior approaches that rely solely on the single most “important” word, our method evaluates the influence of candidate substitutions across the top-p important words to identify the optimal modification. This design inherently accounts for word interactions, as each substitution is assessed within the full sentence context.
> > Our quantitative results in Tables 1–4 support this: the proposed method consistently outperforms baselines that modify only the most salient word, and Table 4 further shows that substituting less important words can produce even stronger adversarial effects—evidence that effectiveness depends on sentence-level interactions rather than isolated word saliency. Visualization results in Figure 1 also illustrate this behavior.
> > Moreover, substitution candidates are generated using a pretrained BERT model, which ensures context-aware, semantically coherent replacements and naturally incorporates word-level dependencies.
> > We will clarify this point in the revised manuscript and provide additional quantitative evidence demonstrating the role of modeling word interactions.
> >
> > 6. Proposition 1
> >
> > Proposition 1 establishes the theoretical foundation for our design by showing that reducing the interaction strength between perturbation units leads to improved transferability. This aligns with observations made in prior empirical studies [1]. We further quantify the interaction values and the theoretical analysis to support it in Section C.1. We will clarify the significance of this proposition in the revised manuscript.
> > [1] Xin Wang, Jie Ren, Shuyun Lin, Xiangming Zhu, Yisen Wang, and Quanshi Zhang. A unified approach to interpreting and boosting adversarial transferability. In Proceedings of the International Conference on Learning Representations, 2020.

---

### Official Review · Reviewer_xW2Z · 2025-10-31

**Soundness:** 3
**Presentation:** 2
**Contribution:** 3
**Rating:** 6
**Confidence:** 4

**Summary:**

In this paper, the authors proposed SEA (Search-space Expansion Attack) to improve the transferability of the generated adversarial text-images pairs from one VLP model (vision-language pretrained model) to others, by introducing more candidates to explore during each optimization step besides the one that's directly chosen by PGD for the image attack and allowing the word replacement to happen at more possible positions instead of only the most saliant word for the text attack. Specifically, the new directions (or informally "gradients") to explore for potential adversarial images are chosen by drawing random linear combinations of all past gradients (difference with last iteration) or past perturbations (difference with original image), and the new word replacement possibilities come from words other than the most saliant one whose change might induce larger drop in text-image similarity. SEA is tested in retrieval, grounding and captioning tasks for different source and target VLP models. It shows improvement relatively significant in retrieval and milder but still consistent in the other tasks comparing to existing attacks, including SA-AET which also aims at improving attack transferability and enlarges the search space to some extent.

**Strengths:**

+ The proposed attack achieves noticeable improvement over existing attacks including very recent ones also working on transferability.
+ The authors focused on expanding the search space and applied the same idea to both the image and text domain.

**Weaknesses:**

+ The writing and presentation are unclear and sometimes even confusing. For instance, while it is understandable to think of the combined gradients as some new "gradient", it is not technically a gradient of anything; symbols like $m\in\{1,2\}$ and $m+$ in Eq. 5 are used without any expiation; Figure 2 doesn't explain at all what the triangular samples are but introduces inverse directions that appear only in this figure an nowhere else for unknown reasons.
+ The linkage between the motivation, the theory and the method are not very strong. For comparison, SA-AET broadens the adversarial image search space by sampling from a triangular region enclosed by original, previous and current compromised images which intuitively defines a search space that is both semantically relevant and more diverse than the neighboring areas of the current image. Dropping the constraints of "regions" is said to be a good thing about SEA but how and why? The proposed linear combination of past gradient or perturbations seem more like a successful trick that the authors picked up purely empirically than designed carefully.

**Questions:**

+ While linearly combining past gradients sounds somewhat understandable, what's the logic behind combining the perturbations? What does $\sum \eta=1$ mean when the norm of the past perturbations are related to the time step?
+ What is the reason for choosing normal distribution for for perturbation-base search space expansion?
+ What selection criterion does Eq. 5 define? What is $m\in\{1,2\}$ and "m+"?
+ How large is $l$ in practice? In the example, "cat" is replaced with "fairy" which doesn't sound very close to each other. Given that TextAttack is the shared library, are the budgets set to the same as your baselines?

---

> ### Author Response · Authors · 2025-11-28
> **Response to Reviewer xW2Z (1)**
>
> We sincerely thank the reviewer for the constructive comments. We address the concerns point-by-point below.
>
> Q1. The logic behind combining the perturbations
>
> The motivation of our method is to expand the adversarial search space and reduce overfitting to a single update direction. To achieve this, we leverage information from past iterations, i.e., gradients or perturbations, to explore a broader set of search directions.
> Although the norm of the past perturbations is related to the time step, as we set small perturbation budget, i.e., 2/255 with step size as step size of 0.5/2.25. After few steps, i.e., 4, the norm of perturbations can achieve the budget, so that the $\sum \eta =1$ is necessary to ensure that the combined update remains within the allowed perturbation budget and prevents uncontrolled growth in magnitude.
>
> Q2.  Normal distribution for perturbation-base search space expansion
>
> Choosing normal distribution aims to explore more search space that uniform distribution cannot explores. As shown in Figure (c) and (d), normal distribution can introduce some inverse update directions, which help to explore more search space.
> To further validate the effectiveness of leveraging both combination strategies, we evaluate variants that use only normal-distributed combinations or only uniform-distributed combinations. The results, shown below, demonstrate that the proposed strategy outperforms either single-distribution variant.
>
> Table 1. Attack success rate $(\%)$regarding the average of R@1 in image-text retrieval on Flickr30K. {*} highlights white-box attack results.
>
> |Target Model|CLIP_ViT-B/16.|CLIP_RN101.|ALBEF. |TCL. |
> | Source Model |I2T |T2I| I2T| T2I. | I2T| T2I.| I2T| T2I. |
> |:---------:|:-----:|:------:|:-----:|:------:|:-----:|:------:|:-----:|:------:|
> |SEA-P w/o uniform-distributed combinations| 99.28*| 98.98*| 51.13| 55.37| 15.87| 31.83| 16.26 |31.27|
> |SEA-P w/o normal -distributed combinations| 99.63 *| 99.58 *| 52.45 | 57.20| 16.79| 32.13| 17.70  |33.34|
> |SEA-P| 99.26*| 99.07*| 53.43| 58.50| 17.62| 33.07| 18.86  |34.69|
>
> Q3.  Selection criterion and notation
>
> The selection criterion in Eq. 5 ensures that newly generated update directions contribute positively to the attack. Specifically, we compute the original loss on the clean image–text pair (i.e., their feature similarity). A candidate perturbation is selected if it increases this loss, meaning it enlarges the image–text dissimilarity. The notation $ m^+$ denotes the resulting set of selected updates, allowing us to distinguish it from the original sets $\mathbb{G}^1_k, \mathbb{G}^2_k, \Delta^{1}_k, \Delta^{2}_k$.
>
> Q4. Text attack
>
> 4.1. Following classical methods in the field [1–2], we set $l =10$.
>
> 4.2. Consistent with prior work [1–3], the proposed method adopts context-aware word substitutions, where substitution candidates are predicted based on the surrounding context. We further apply a semantic-similarity threshold (0.3) to filter out dissimilar words. Although “cat’’ and “fairy’’ may not seem closely related in isolation, their substitution does not substantially alter the overall sentence-level meaning, which is standard practice in current text-attack settings.
>
> 4.3. We set the same budget as the compared methods.

---

> > ### Author Response · Authors · 2025-11-28
> > **Response to Reviewer xW2Z (2)**
> >
> > W1. The writing and presentation
> >
> > Thanks for pointing out, which is very important for improving the clarity of the paper.
> > The notation $m \in 1,2$ denotes the sets generated by gradient-based and perturbation-based search space expansion, respectively, while $m^+$ refers to the selected subset after applying our selection criterion.
> > In Figure 2, the triangular markers represent newly generated adversarial examples produced by our search-space expansion strategies. The inverse directions observed in panels (c) and (d) arise from the use of different combinations of historical update information sampled from a normal distribution, which introduces negative values and therefore produces direction reversal. This design intentionally broadens the range of explored directions, enabling more effective search and improved attack performance.
> >
> > W2. Comparison with SA-AET and linkage among motivation, theory, and method.
> >
> > SA-AET expands the image search space through geometric sampling within a triangular region, selecting new samples that preserve attack performance while increasing diversity.
> > In contrast, our method focuses on enhancing diversity and reducing correlation among update directions by expanding the search space in a more flexible manner. We further introduce a selection mechanism to ensure that the newly generated gradients or perturbations remain effective for the attack, thereby maintaining semantic relevance within the expanded search region.
> > The combination of different perturbations is not an empirical trick. Rather than assigning predefined weights, we employ a principled selection process to identify effective combinations. Experiments show that expanding the search space can enhance transferability.
> > In addition, we provide a theoretical analysis from the perspective of perturbation-unit interactions, which further supports the design of our method. Together, these components establish a coherent link between the motivation, theoretical foundation, and methodological design of the proposed approach.
> > We will better emphasize this motivation–theory–method linkage in the revised manuscript.
> >
> > [1] Set-level guidance attack: boosting adversarial transferability of vision-language pre-training models
> >
> > [2] VLATTACK: Multimodal Adversarial Attacks on Vision-Language Tasks via Pre-trained Models
> >
> > [3] Bert-attack: adversarial attack against bert using bert

---

### Official Review · Reviewer_tWKh · 2025-10-31

**Soundness:** 3
**Presentation:** 2
**Contribution:** 3
**Rating:** 4
**Confidence:** 4

**Summary:**

This paper proposes SEA (Search-space Expansion Attack), a unified framework that improves cross-model transferability by enlarging the adversarial search space across both modalities. For images, SEA leverages historical updates to explore novel optimization directions, effectively avoiding suboptimal optimization trajectories and overfitting. For text, SEA considers both individual-word importance and word interactions, recognizing that less salient words can sometimes yield stronger, more transferable attacks.
However, I found that some of the paper’s claims are not substantiated and there are several noticeable grammatical errors; therefore, I believe the manuscript should be thoroughly revised before being considered for acceptance.

**Strengths:**

1. This paper is relatively complete.
2. An interesting point is the enlargement of the adversarial search space across both text and image modalities.

**Weaknesses:**

1. There are some typos in this paper. For example, (1) In Equation (6), the summation is written as $\hat{\Delta }_{t}$. (2) Table 2's caption: "isual". (3) In Section 3.3.1, the author refers to “Figure 4”, but this figure does not appear in the paper.

2. The paper claims that using historical update information can avoid overfitting and local optima, but no empirical or theoretical evidence is provided to substantiate this claim.

3. The motivation highlights factors affecting text presentation—namely the semantics of individual words, substitution candidates, and their contextual relationships. But the proposed text attack does not explicitly model or target these factors.

**Questions:**

1. What are the actual memory usage and runtime speed of the proposed method?

2. Please refer to the weaknesses section.

---

> ### Author Response · Authors · 2025-11-28
> **Response to Reviewer tWKh**
>
> We sincerely thank the reviewer for the constructive comments. We address the concerns point-by-point below.
>
> W1. Typos
>
> Thanks for pointing them. We use $\tilde{\Delta}_k$ to indicate the set that contains three kinds of perturbations.  Figure 4 actually refers to Table 4. We will correct them in the revised version.
>
> W2. Historical update information
>
> Overfitting refers to the case that the attacker drops into local optima and thus overfit to the source model. The motivation of our method is to expand the adversarial search space to reduce overfitting. To improve computational efficiently, we reuse available historical information, i.e., past gradients. To verify the effectiveness of the proposed strategy, we conduct both empirical experiments and theoretical analysis.
> Empirical experiments: For example, in Table 1, 2, and 3, all methods perform well in the white-box settings, while their performance degrades in black-box cases, i.e., attacking other models on the same tasks and other tasks.  In comparison, the proposed method can achieve better performance than compared methods, which demonstrates that the proposed method can effectively avoid overfitting.
> Theoretical analysis: To further verify the effectiveness of the proposed method, we conduct the theoretical analysis from another complementary perspective in Sec. 2.3.1. It verifies that the proposed method can decrease the interactions between perturbation units. We also provide he interaction values of perturbations generated by different attacks, demonstrating the effectiveness of the proposed method in Sec C.1.
>
> W3. factors affecting text presentation
>
> The proposed method does consider the factors affecting text presentation that the reviewer mentioned. Instead of relying solely on the most “important” word, we evaluate the actual influence of substitutions on the top-p important words on cross-modal alignment to discover the optimal substitution. This design explicitly incorporates both (a) the semantics of individual words and (b) the semantics of substitution candidates, and  (c) contextual relationships. Our experimental results in Tables 1–4 and visualization results in Figure 1 support this.
> From experimental results in Table 1, 2, and 3, it can be observed that the proposed method can achieve better performance than compared methods that substitutes only the most important word. In particular, in Table 4, we also measure the influence of p, which shows that considering replacements on other less important words would achieve better performance than only considering the most important word. This demonstrates that text perturbation effectiveness depends on more than just word-level importance. It also depends on the semantic properties of substitution candidates and their interactions with the overall sentence.
> This phenomenon is further illustrated in Figure 1: although “fence” is ranked more important than “run,” modifying the less important word “run” still leads to stronger cross-modal disruption than BERT-Attack. This shows that our method captures factors beyond per-word importance and benefits from exploring a broader perturbation space.
> Finally, substitution candidates are generated using a pretrained BERT model, ensuring context-aware replacements.
> In summary, our method does account for the semantic and contextual factors influencing text representation, and the empirical results clearly demonstrate its effectiveness.
>
> Q1. Memory usage and runtime speed
>
> As shown in Table 4, when taking CLIP_{ViT-B/16} as the source model, the run time per sample is 3.07s and 0.199s for image and text respectively. The GPU memory usage is 7221MB. In comparison, the compared method, e.g., SGA method, takes 2.38s for images and 0.19s for the texts, while the GPU memory usage is 7117 MB. Give the performance gains, the efficiency of the proposed method is acceptable. We will reflect this is our revised version.

---

### Official Review · Reviewer_hbij · 2025-10-31

**Soundness:** 2
**Presentation:** 3
**Contribution:** 2
**Rating:** 4
**Confidence:** 5

**Summary:**

This paper proposes an attack named SEA (Search-space Expansion Attack) to improve the transferability of adversarial attacks across vision-language pre-trained (VLP) models. SEA expands the adversarial search space in both image and text modalities. It leverages historical gradients for image perturbations and performs multi-word substitutions that account for both word importance and interactions in text. The authors conduct extensive experiments across various VLP models and tasks, showing consistent but modest improvements over existing approaches, accompanied by solid theoretical analysis.

**Strengths:**

1. Quality: The theoretical and empirical analyses on transferability are rigorous and add credibility to the findings.

2. Clarity: The paper is clearly written and well-organized, making the proposed framework easy to understand.

3. Significance: While the performance gain is moderate, the paper contributes another aspect of insights into improving cross-modal adversarial transferability, an important aspect of robustness evaluation. The approach is methodically implemented and systematically evaluated across diverse settings.

**Weaknesses:**

1. Incremental contribution: The main ideas of expanding the search space and using historical updates are extensions of well-studied techniques in adversarial optimization. The novelty is limited.

2. Minor empirical improvement: The performance gains in Table 1 are relatively small, suggesting that the proposed method offers incremental progress rather than a clear leap forward.

3. Incomplete related work discussion: The discussion of textual adversarial attacks is brief and omits important gradient-based approaches such as "LeapAttack: Hard-Label Adversarial Attack on Text via Gradient-Based Optimization", which are directly relevant.

4. Limited insight on cross-modal interactions: While SEA aims to expand search space across modalities, the analysis could further explain how image and text perturbations jointly enhance transferability.

**Questions:**

1. Since the improvement is modest, could the authors further clarify the necessity of this type of method?
2. Would integrating recent hard-label text attack baselines (e.g., TextHoaxer and LeapAttack) change the comparative performance results?
3. What's the performance for attacking the latest VLP models such as InternVL and Qwen-VL?

---

> ### Author Response · Authors · 2025-11-28
> **Response to Reviewer hbij**
>
> We sincerely thank the reviewer for the constructive comments. We address the concerns point-by-point below.
> 1. Novelty
> The motivation of our method is to expand the adversarial search space to reduce overfitting. We acknowledge that some existing methods also leverage historical updates, such as momentum. However, the proposed method is substantially different. Momentum simply accumulates a moving average of past gradients to stabilize the current update direction. In contrast, our method explicitly explores diverse search directions to prevent overfitting in adversarial optimization. To improve efficiency and practicability of the proposed method, we reuse available historical information, i.e., past gradients, but the key lies in expanding the search space instead of using historical information. In Table 4, we demonstrate that increasing the diversity of search directions can enhance adversarial transferability.
>
> 2. Minor empirical improvement
> SEA demonstrates consistent and notable improvements across all evaluated architectures, with especially strong gains in the black-box cross-architecture scenario. Overall, SEA-P and SEA-G surpass the strongest baseline (SA-AET) by an average of 11.51%, highlighting the effectiveness of our search-space expansion strategy.
>
> 3. Incomplete related work discussion
> Thanks for the suggestions, we will make more thorough reviews on related methods in the revised version to help readers better understand the filed, including textual adversarial attacks [1-2].
>
> 4. Limited insight on cross-modal interactions
> Our method explicitly enhances cross-modal disruption in several ways:
> SEA explicitly disrupts cross-modal alignment through both image- and text-side search space expansion.
>
> 4.1. Image-side expansion produces diverse adversarial examples that alter visual features along multiple directions, enabling broader exploration of cross-modal similarity landscapes.
>
> 4.2. Text-side expansion measures the effect of substituting influential words on the image–text similarity score, thereby generating diverse image–text pairs that directly probe cross-modal associations.
>
> Together, these expansions explore cross-modal interactions more comprehensively and improve transferability, consistent with observations in prior work [3]. We will clarify this more explicitly in the revised version.
>
> [1] LeapAttack: Hard-Label Adversarial Attack on Text via Gradient-Based Optimization
> [2] TextHoaxer: Budgeted Hard-Label Adversarial Attacks on Text
> [3] VLATTACK: Multimodal Adversarial Attacks on Vision-Language Tasks via Pre-trained Models

---

### Meta-Review · Area_Chair_NYDW · 2026-01-07

**Summary:**

This paper proposes SEA, a unified framework that improves cross-model transferability by enlarging the adversarial search space across both modalities. However, the reviewers have raised various concerns and the authors have failed to adequately address the fundamental problems regarding the paper's novelty, methodological rigor, and experimental validation.

Reviewer hbij's concerns about incremental contribution and limited performance improvements remained unaddressed, as the authors only highlighted gains in a single scenario while neglecting most other settings and failing to integrate recent baselines. Reviewer tWKh's request for direct experimental evidence on overfitting reduction and local optima escape remained unmet, with the authors providing only attack results that could be attributed to various other factors rather than the claimed mechanisms.

Reviewer xW2Z's key questions about the removal of the "region" constraint and the foundation of the linear combination approach were met with vague explanations lacking specific details or direct experiments.

Reviewer b2fs's concerns about unquantified component contributions and unverified optimization claims persisted, as the authors did not provide direct evidence of avoiding local optima or quantitative indicators for text interactions.

Moreover, throughout the discussion, the authors have not addressed all three questions raised by Reviewer hbij.

Considering the above factors, we believe that the current version of this paper is far from meeting the acceptance standards of a top-tier AI conference like ICLR.

**Reviewer Concerns:**

- Reviewer hbij pointed out the incremental contribution, minor empirical improvements, incomplete related work discussion, and limited insight on cross-modal interactions of this submission. The authors supplemented little discussion on related work, which only partially addressed the concern of incomplete related work discussion. The authors' rebuttal regarding novelty remained confined to emphasizing the use of historical information and search space, which was still based on existing techniques as Reviewer hbij had pointed out. Regarding the limited performance improvements, the authors only highlighted the significant gains in the black-box cross-architecture scenario, while the method's performance improvements in most other scenarios remained limited. The authors' response to cross-modal interactions was still limited to the mechanisms within each modality, lacking discussion on how synergistic interactions between modalities enhance transferability. Moreover, the questions (necessity of this method, integrating more recent baselines, performance on the latest VLP models such as InternVL and Qwen-VL) raised by Reviewer hbij were completely neglected by the authors. Thus, the authors' professionalism is questionable.
- Reviewer tWKh pointed out multiple typos, unsupported claims regarding historical updates, and a misalignment between the motivation and the proposed method, and requested the results on memory usage and runtime speed. The authors have fixed the typos and provided the results on memory usage and runtime speed. However, the authors' explanation regarding the historical update information remained limited to the attack effectiveness of the method (Table 1, 2, and 3), without directly providing experimental evidence on the reduction of overfitting and local optima. The attack effectiveness could be attributed to various other factors, and it cannot be definitively concluded that it was necessarily due to the reduction of overfitting and local optima. The authors should present more direct and clear experimental results. Furthermore, the authors' discussion on factors affecting text presentation still failed to clarify how the method utilizes the semantics of individual words, substitution candidates, and their contextual relationships. The provided experimental results remained confined to the attack effectiveness of the method, rather than being direct experimental evidence, and lacked quantitative indicators related to text semantics.
- Reviewer xW2Z pointed out the unclear and confusing writing, the weak linkage between the motivation, theory, and method, and raised various questions. The authors have tried to clarify these issues. However, the reviewer raised two key concerns: (1) why the "region" constraint was removed (a key difference from SA-AET for this work) and how this constitutes the advantage of SEA; (2) whether the proposed linear combination of past gradients/perturbations is a carefully designed component rather than a purely empirical trick. Unfortunately, the authors' response provided only vague explanations, such as "principled selection process" and "theoretical analysis from the perspective of perturbation-unit interactions" without offering specific details or direct experiments to answer these questions. They did not explicitly explain how abandoning the region constraint enhances diversity while preserving semantic relevance (i.e., the "how and why" emphasized by the reviewer), nor did they elaborate on the specific theoretical foundation or design principles underlying the linear combination to distinguish it from empirical heuristics. The mention of experimental results and theoretical analysis lacks sufficient depth to bridge the gap between motivation, theory, and methodology, leaving the logical chain still broken.
- Reviewer b2fs pointed out the unquantified component contributions, unverified optimization claims, absent result visualizations, missing methodological framework, unsupported interaction claims, and unclear proposition significance of this submission. However, the authors' explanation regarding local optima is still limited to the attack effectiveness (Table 1, 2, and 3), which may stem from various other factors rather than the local optima. It cannot directly prove whether SEA truly escapes local optima or merely converges to different ones. Additionally, the authors did not provide direct experimental evidence to demonstrate that they have leveraged word interactions. They only presented the attack outcomes (Table 1-4), lacking quantitative indicators for interactions between texts, as well as experiments showing how the extent of such interactions influences the method's performance.

**Reviewer Scores:**

- Reviewer hbij: As mentioned before, most of the concerns from Reviewer hbij remained unaddressed. So I believe that Reviewer hbij will keep the score (4).
- Reviewer tWKh: As mentioned before, most of the concerns from Reviewer tWKh remained unaddressed. So I believe that Reviewer tWKh will keep the score (4).
- Reviewer xW2Z: As mentioned before, some of the concerns from Reviewer xW2Z remained unaddressed. So I believe that Reviewer xW2Z will keep the score (6).
- Reviewer b2fs: As mentioned before, most of the concerns from Reviewer b2fs remained unaddressed. So I believe that Reviewer b2fs will keep the score (2).

---

### Decision · Program_Chairs · 2026-01-26

Reject